# GramSeq-DTA: A Grammar-Based Drug–Target Affinity Prediction Approach Fusing Gene Expression Information

**DOI:** 10.3390/biom15030405

**Published:** 2025-03-12

**Authors:** Kusal Debnath, Pratip Rana, Preetam Ghosh

**Affiliations:** 1Department of Computer Science, Virginia Commonwealth University, Richmond, VA 23284, USA; debnathk@vcu.edu; 2Department of Computer Science, Old Dominion University, Norfolk, VA 23529, USA; prana@odu.edu

**Keywords:** drug–target affinity, deep learning, grammar-based encoding, chemical perturbation, multi-modal

## Abstract

Drug–target affinity (DTA) prediction is a critical aspect of drug discovery. The meaningful representation of drugs and targets is crucial for accurate prediction. Using 1D string-based representations for drugs and targets is a common approach that has demonstrated good results in drug–target affinity prediction. However, these approach lacks information on the relative position of the atoms and bonds. To address this limitation, graph-based representations have been used to some extent. However, solely considering the structural aspect of drugs and targets may be insufficient for accurate DTA prediction. Integrating the functional aspect of these drugs at the genetic level can enhance the prediction capability of the models. To fill this gap, we propose GramSeq-DTA, which integrates chemical perturbation information with the structural information of drugs and targets. We applied a Grammar Variational Autoencoder (GVAE) for drug feature extraction and utilized two different approaches for protein feature extraction as follows: a Convolutional Neural Network (CNN) and a Recurrent Neural Network (RNN). The chemical perturbation data are obtained from the L1000 project, which provides information on the up-regulation and down-regulation of genes caused by selected drugs. This chemical perturbation information is processed, and a compact dataset is prepared, serving as the functional feature set of the drugs. By integrating the drug, gene, and target features in the model, our approach outperforms the current state-of-the-art DTA prediction models when validated on widely used DTA datasets (BindingDB, Davis, and KIBA). This work provides a novel and practical approach to DTA prediction by merging the structural and functional aspects of biological entities, and it encourages further research in multi-modal DTA prediction.

## 1. Introduction

Drug–target affinity (DTA) prediction provides a foundation for modern drug discovery, with various benefits that improve efficiency, reduce costs, and increase success rates. Affinity reflects the likelihood of interaction of a drug–target pair, typically measured by
$K_{d}$ (dissociation constant) or
$K_{i}$ (inhibition constant). Strong binding is indicated by low
$K_{d}$/
$K_{i}$ (nM to pM), while weak binding corresponds to high
$K_{d}$/
$K_{i}$ (
μM to mM). The significance of DTA prediction is well-discussed in the literature, emphasizing its role in accelerating the identification of potential drug candidates and minimizing the risk of failure during clinical trials [1,2]. Recent improvements in computational methods [3,4,5,6] and the availability of relevant data enhance the accuracy and reliability of DTA predictions [7,8], aiding in the design of efficient therapeutic strategies and effective treatments for many diseases.

To achieve accurate DTA predictions, the way in which both drugs and targets are represented is a critical determinant of the performance of the models. The proper encoding of these molecular entities is essential for capturing the intricate relationships between their structural and functional properties. Early computational approaches to DTA often relied on simplified representations, such as molecular fingerprints for drugs and amino acid sequences for proteins, to feed machine learning models. Although these methods showed some promise, their inability to fully capture the complexity of molecular interactions limited the predictive power of DTA models [9,10,11]. As the field evolved, researchers began exploring more sophisticated techniques to better model the structural and chemical properties of drugs and targets, leading to the development of advanced representation methods that significantly improved the predictive accuracy and generalizability of DTA models [12,13,14].

Ozturk et al. [15] proposed DeepDTA, where they utilized 1D convolutional neural networks (CNNs) to extract high-level representations of protein sequences and 1D SMILES representations of the compounds. Before this approach, most computational methods treated drug–target affinity prediction as a binary classification problem. DeepDTA redefined the problem as a continuum of binding strength values, providing a broader view of drug–target interactions.

Nguyen et al. [16] advanced the field by representing drugs as graphs instead of linear strings and utilized graph neural networks (GNNs) to predict drug–target affinity in their proposed deep learning model called GraphDTA. Building on this trend, Tran et al. [17] proposed the Deep Neural Computation (DeepNC) model, which consists of multiple graph neural network algorithms.

The rise of natural language-based methods in biomedical research has led to further innovations in DTA modeling [18,19,20,21]. Qiu et al. [22] introduced G-K-BertDTA to bridge the gap between the structural and semantic information of molecules. In their approach, drugs were represented as graphs to learn their topological features, and a knowledge-based BERT model was incorporated to obtain the semantic embeddings of the structures, thereby enhancing feature information.

Nevertheless, there are some limitations to the above-mentioned approaches. Firstly, the information on the relative positions of the constituent atoms and bonds is often missing in the drug encoding approaches adopted in these models. In addition, the functional aspects of those drugs, which can provide relevant insights into their interaction with targets, were also not incorporated.

To address these limitations, we utilized the encoding approach for drugs known as the grammar variational autoencoder (GVAE) proposed by Kusner et al. [23]. The GVAE discusses the parse tree-based encoding of the drug entities, which allows learning from semantic properties and syntactic rules. This approach can learn a more consistent latent space representation in which entities with nearby representations decode to similar outputs. In addition, to incorporate the functional aspect of those drugs, we integrated the chemical perturbation information from the L1000 project [24]. In the L1000 project, various chemical entities have been used as perturbagens and were tested against multiple human cell lines, primarily linked to several types of cancers, to analyze their gene expression profile. We have utilized these gene expression signatures as the functional feature set for the drugs. Thus, the approach followed in this work utilizes the structural and functional representation of drugs, which enhances the drug–target affinity prediction and outperforms the current state-of-the-art methods.

The paper provides a thorough overview of the background research in Section 2, along with a detailed explanation of the methodology, dataset preparation, network architecture, and evaluation metrics in Section 3. Section 4 presents the results of the proposed method on commonly used benchmark datasets and compares its performance to the current state-of-the-art DTA prediction models. Finally, Section 5 addresses the limitations of the study and explores opportunities for future research advancements.

## 2. Background

### 2.1. Grammar Variational Autencoder

Gómez-Bombarelli et al. [25] used Gated Recurrent Units (GRUs) and Deep Convolutional Neural Networks (DCNNs) to develop a generative model for molecular entities based on SMILES strings. This model has the potential to encode and decode molecular entities through a continuous latent space, which aids in the exploration of novel molecules with desirable properties in this space. Nevertheless, one major drawback in using string-based representation for molecular entities is their fragility, i.e., minute alterations in the string can lead to a complete deviation from the original molecule, even corresponding to the generation of entirely invalid entities. James et al. [26] first proposed the concept of constructing grammar for chemical structures. According to this work, every valid discrete entity can be represented as a parse tree from the given grammar. The advantage of generating parse trees compared to texts is that it ensures the complete validity of the generated entities based on grammar. Thus, Kusner et al. [23] proposed the grammar variational autoencoder (GVAE), which encodes and decodes directly from these parse trees. This approach allows the GVAE to learn from syntactic rules as well as to learn semantic properties. Along with its ability to generate valid outputs, this approach can also learn a more coherent latent space representation in which entities with nearby representations decode to similar outputs.

#### 2.1.1. Context-Free Grammar

Context-free grammar (CFG) is conventionally defined as a 4-tuple *G =* (*V, Σ, R, S*), where *V* represents a finite set of non-terminal symbols;
Σ represents a finite set of terminal symbols, disjoint from *V*; *R* represents a finite set of production rules; *S* is a unique non-terminal referred to as the start symbol; and *G* represents the grammar that describes a set of trees that can be formed by applying rules in *R* to leaf nodes until all leaf nodes become terminal symbols in
Σ.

The rules *R* are technically defined as
α→βforα∈Vandβ∈(V∪Σ*), * denoting the Kleene closure. Practically, these rules are portrayed as a collection of mappings from a solitary non-terminal on the left-hand side in *V* to a sequence of symbols that can be either terminal or non-terminal by definition. These mappings can be seen as a rule for rewriting.

When a production rule is applied to a non-terminal symbol, it creates a tree structure where the symbols on the right-hand side of the production rule become child nodes for the left-hand side parent. These trees extend from each non-terminal symbol in *V*. The language of *G* is the set of all sequences of terminal symbols that can be generated by traversing the leaf nodes of a tree from left to right. A parse tree is a tree with its root at *S* and a sequence of terminal symbols as its leaf nodes. The prevalence of context-free languages in computer science is attributed, in part, to the existence of practical parsing algorithms.

#### 2.1.2. Syntactic vs. Semantic Validity

A crucial aspect of grammar-based encoding is that the encoded molecules are syntactically valid, but the semantic validity of these molecules is a matter of discussion. There are several reasons behind this phenomenon as follows: (a) Some molecules produced by the grammar may be unstable or chemically invalid; for example, a carbon atom cannot make bonds with more than four atoms in a molecule as it has a valency of 4. Nevertheless, the grammar can produce this kind of molecule. (b) The assignment of ring-bond digits in SMILES is a non-context-free process. It needs to keep track of the order in which rings are encountered, along with the connectivity between the rings, which can only be determined from the local context of the string. For example, in naphthalene (c1ccc2c(c1)cccc2), the outer ring uses the digit ‘1’, and the inner ring uses ‘2’. The digits are not nested but rather follow a specific order. (c) The GVAE can still produce an undetermined sequence if there are existing non-terminal symbols on the stack after processing all logit vectors.

### 2.2. L1000 Assay

The L1000 project [24] is part of the Library of Integrated Network-Based Cellular Signatures (LINCS) program funded by the National Institutes of Health (NIH). This program aims to catalog and analyze cellular responses to various perturbations to understand how these perturbations modulate cellular functions. This project efficiently manages chemical perturbagen data using a structured method, involving data generation, processing, and analysis. This project uses various chemical compounds, including FDA-approved drugs, experimental drugs, natural compounds, and other bioactive molecules, as perturbagens. Perturbagen selection mainly involves the possible connection of these chemicals to numerous biological pathways and disease cross-talks. Various human cell lines, primarily associated with several types of cancers, are chosen to ensure diversity in biological responses. The L1000 data can thus be used to identify potential new uses for existing drugs or to discover new candidate drugs. Moreover, novel hypotheses can be made that correspond to the possible effects of the new compounds by performing comparative analyses of the gene expression signatures of known drugs. The data also aid in understanding the underlying molecular mechanism of the diseases by showcasing the effect of different compounds in the alteration of gene expression related to disease pathways.

## 3. Materials and Methods

### 3.1. Datasets

#### 3.1.1. Benchmark Datasets

In this study, the following three datasets are used for benchmarking purposes: BindingDB [27], Davis [28], and KIBA [29]. The drug–target affinity dataset in the BindingDB database contains experimental binding affinities between small molecules and protein targets and pharmacological annotations of the entities (e.g., ID, Structure, etc.). In the Davis dataset, the targets are kinase proteins, and the drugs are the small molecules (inhibitors) targeting those kinases. Similar to Davis, the KIBA dataset also focuses on kinase proteins and their corresponding inhibitor drugs, but it contains a more significant number of instances compared to Davis. The variations in drug and target components across these datasets, thus, allow for a comprehensive evaluation of the robustness and generalizability of the model.

In these datasets, the drugs are represented as SMILES strings, and the target proteins are represented as amino acid sequences. For the BindingDB and Davis datasets, the labels are the
$K_{d}$ (dissociation constant) values, which indicate the extent of the interactions between each drug–target pair. Meanwhile, a unified KIBA score is used as a label for the KIBA dataset, combining
$K_{d}$,
$K_{i}$ (inhibition constant), and
$IC_{50}$ (half-maximal inhibitory coefficient) values for the corresponding drug–target pairs. While Davis and KIBA are readily available for training without any prior preprocessing, BindingDB contains the missing values among various features. Thus, thorough preprocessing is performed on BindingDB, keeping only the SMILES target protein sequences and the
$K_{d}$ values as the required features, discarding everything else. Even if this preprocessing leads to a certain amount of data loss, this step eventually produces cleaner data suitable for model training. The labels are converted into logarithmic form which helps improve the performance of the model in regression tasks.

#### 3.1.2. L1000 Chemical Perturbation Dataset

The chemical perturbation data available in the L1000 project is documented in raw format. Therefore, the data need to be processed accordingly for use. The detailed process of preparing the L1000 dataset from the raw data is discussed below.

① The L1000 chemical perturbation data file is loaded where each perturbagen has multiple replicates based on dosage concentration, and each replicate has two lists of associated genes—one for up-regulated and the other for down-regulated genes. ② The analysis of the dosage concentration distribution among the replicates shows that samples with a concentration of
10μM are the most common. Gene expression responses can be dose-dependent; thus, selecting single-dose samples provides more biological relevance, providing a standardized functional overview of the drug effects. Therefore, for standardization purposes, samples with a concentration of
10μM are selected for further analysis, while the others are excluded. ③ Each unique perturbagen is mapped into its corresponding SMILES representation, which is important for downstream molecular modeling. ④ For each perturbagen
$p_{i}$, the gene regulatory information is represented as a vector of ‘up’- and ‘down’-regulation values across 978 landmark genes, and the number of times a gene is up-regulated or down-regulated is counted and normalized by the number of replicates. Let
$x_{ij}^{up}$ represent whether gene *j* is up-regulated for perturbagen
$p_{i}$ and
$x_{ij}^{down}$ represent the same for down-regulation. The final regulatory vector for each perturbagen is computed as follows:
(1)$$v_{i}=\frac{1}{\mathrm{count}(p_{i})}\left(\sum_{j=1}^{m}x_{ij}^{\mathrm{up}},\sum_{j=1}^{m}x_{ij}^{\mathrm{down}}\right)$$ where
m=978 is the number of landmark genes, and
$\mathrm{count}(p_{i})$ is the number of times perturbagen
$p_{i}$ appears in the dataset. A representative illustration is shown in Figure 1. ⑤ Finally, the vector for each perturbagen is stacked to obtain the final matrix as follows:
(2)$$V={\left\{v_{i}\right\}}_{i=1}^{k},\;v_{i}\in R^{978\times 2}$$ where
V is of shape
k×978×2, and *k* is the number of unique perturbagens. The processed dataset is an important contribution of this work that can aid future research in RNA-Seq data integration and analysis.

The results from the L1000 assay highlight the potential to capture any cellular state by measuring the reduced representation of the human transcriptome. In that study, 12,031 Affymetrix HGU133A expression profiles from the Gene Expression Omnibus (GEO) [30] were analyzed. The analysis determined that the optimal number of informative transcripts (k) is 978. Selecting a lower value for k results in significant information loss, while higher values fail to offer a substantial cost reduction compared to the entire transcriptome. Ultimately, these 978 selected landmarks were sufficient to represent 82% of the complete transcriptomic information. Thus, for the construction of the chemical perturbation dataset here, that list of 978 genes was used.

Not all the perturbagens mentioned in the L1000 dataset are entirely present in the benchmark datasets. Therefore, the datasets are processed accordingly, and only those drugs whose corresponding regulatory vector is present in the L1000 dataset are selected. The processing of the datasets resulted in a decrease in the number of total interactions. A summary of all the original and processed benchmark datasets is presented in Table 1.

### 3.2. Network Architecture

The complete network consists of four main parts: (a) a drug encoder, (b) RNA-Seq encoder, (c) protein encoder and (d) regression head. Figure 2 shows the schematic diagram of the complete network architecture.

#### 3.2.1. Drug Encoder

For this work, we utilized a pre-trained GVAE model from the study conducted by Zhu et al. [31] that focuses on deep learning-based drug efficacy prediction from transcriptional profiles. One-hot encoded vectors are generated by parsing the SMILESs representations of the drugs using a grammar-tree-based approach and subsequently passing those into the encoder network. The detailed process of parsing the SMILES and generating one-hot encoded vectors is discussed below.

① RDKit [32] tool is used to canonize all SMILESs to a standardized representation of SMILESs, as the same molecule collected from different sources may possess different SMILES representations. ② Canonized SMILES representations are converted into a collection of tokens using a tokenizer. ③ The tokenized sequence is then parsed using a grammar adopted from the work of Kusner et al. [23]. This yields a sequence of production rules as follows:
(3)$$G\left(\tau \left(S\right)\right)=P=\left\{P_{1},P_{2},\ldots ,P_{q}\right\}$$ where *G* is the grammar,
τ(S) is the tokenized sequence, and
$P=\{P_{1},P_{2},\ldots ,P_{q}\}$ is the sequence of production rules. Each production rule is then mapped to an index
$I_{i}$ in a predefined list of rules. ④ A zero matrix is initialized, denoting the vector to be populated by one-hot encoding as follows:
(4)$$O_{j,I_{j}}=1,\;\forall j\in \left\{1,2,\ldots ,\min\left(M,q\right)\right\},\;I_{j}\in \left\{1,2,\ldots ,N-1\right\}$$ where *O* is the encoded one-hot vector of shape
M×N, *M* is the maximum length of sequences, N is the total number of production rules, and *q* is the number of productions. If q is smaller than M, the rest of the matrix is padded with an indicator for “end of sequence”, as follows:
(5)$$O_{j,N-1}=1,\;\forall j\in \left\{q+1,q+2,\ldots ,M\right\}$$

In this work, the values of *M* and *N* are 277 and 76, respectively. The generated one-hot vectors for each SMILES representations are then passed into the encoder network. The schematic diagram of the overall process is given in Figure 3.

#### 3.2.2. RNA-Seq Encoder

A fully connected neural network (FCNN) [33] is used for the extraction of meaningful features from the high-dimensional RNA-Seq data. As discussed in the L1000 chemical perturbation dataset preparation, the resulting dataset is of
k×978×2 shape, where *k* is the number of unique perturbagens, 978 is the number of landmark genes, and 2 indicates the number of columns representing up-regulated and down-regulated genes. When these vectors are passed through a fully connected neural network, it learns a condensed and abstract representation of how each perturbagen affects gene expression.

#### 3.2.3. Protein Encoder

Feature extraction from amino acid sequences is achieved using two different types of neural networks as follows: a Convolutional Neural Network (CNN) and a Recurrent Neural Network (RNN) [33]. CNNs are able to identify local motifs and patterns within a sequence by using the sliding windows of filters to capture neighboring amino acids. On the other hand, RNNs are capable of extracting long-range dependencies and sequential relationships between amino acids by retaining information from previous positions in the sequence in a step-by-step manner. To encode the sequences, a dictionary of all possible amino acid symbols in the proteins is created. One-hot encoding of a given protein is carried out based on the presence of a particular amino acid in that protein. For standardization, the maximum length of a protein is limited to 1000 sequences. The encoding of all the proteins results in the creation of a tensor of
p×26×1000, where *p* is the number of unique proteins and 26 is the length of the amino acid dictionary.

### 3.3. Training Settings

A pre-trained GVAE model from Zhu et al. [31] is used as the drug encoder for deep learning-based drug efficacy prediction from transcriptional profiles. Since the model is already pre-trained, no additional training is needed for this encoder block. It generates a latent representation for the drugs, which is then concatenated with representations extracted separately—by the FCNN block for RNA-Seq data and by the CNN and RNN blocks for protein data. Finally, these combined representations are fed into a regression head, consisting of fully connected layers, to predict affinity. Unlike the drug encoder, the RNA-Seq and protein encoders, as well as the regression head, are trained from scratch. The combined network is trained together in an end-to-end manner. The training process is set to run for 500 epochs with an adaptive learning rate that starts at 0.001 using the Adam optimizer. The higher learning rate value is chosen to ensure that the training process does not significantly impact the pre-trained weights in the GVAE model. A batch size of 256 has been found to yield the best results, maintaining a balance between memory usage and convergence speed. A summary of the overall network architecture and training hyperparameters are presented in Table 2.

### 3.4. Evaluation Metrics

Evaluating deep learning models involves various metrics that capture different aspects of the performance. Mean Squared Error (MSE) measures the average squared difference between the actual and the predicted values, highlighting prediction accuracy. We also used the Concordance Index (CI), which is a preferred metric for the survival analysis, to evaluate the consistency between the predicted risk scores and the actual outcomes. Together, these metrics provide a robust framework for model evaluation.

#### 3.4.1. Mean Squared Error (MSE)

The Mean Squared Error (MSE) is a common loss function used for regression tasks. It measures the average of the squares of the errors, which are the differences between the predicted and actual values. Mathematically, it is defined as follows:
(6)$$\mathrm{MSE}=\frac{1}{n}\sum_{i=1}^{n}{\left(y_{i}-{\hat{y}}_{i}\right)}^{2}$$ For datasets BindingDB and Davis,
$y_{i}$ is the actual negative log of the
$K_{d}$ value,
${\hat{y}}_{i}$ is the predicted negative log of the
$K_{d}$ value, while
$K_{d}$ is replaced by the KIBA score for the KIBA dataset. *n* is the number of data points.

#### 3.4.2. Concordance Index (CI)

The Concordance Index (CI) is a metric used primarily in survival analysis to evaluate the predictive accuracy of risk scores. It assesses the degree of concordance between the predicted and actual ordering of event times. The CI is able to measure how often a randomly chosen pair is ranked correctly with respect to the ground truth, which is particularly relevant for ranking-based problems like drug–target affinity prediction. The CI is calculated as follows:
(7)$$CI=\frac{\mathrm{Number}\;\mathrm{of}\;\mathrm{concordant}\;\mathrm{pairs}}{\mathrm{Number}\;\mathrm{of}\;\mathrm{possible}\;\mathrm{evaluation}\;\mathrm{pairs}}$$ A pair is considered concordant if the predicted and actual orderings of two instances are consistent.

## 4. Results and Discussion

In this section, we will discuss the performance of the GramSeq-DTA model in detail. The discussion can be divided into the following parts: (a) benchmarking the performance of GramSeq-DTA with integrated RNA-Seq information against the baseline models, (b) the advantage of integrating RNA-Seq information to the model, and (c) performance comparison of the proposed model using original and processed datasets.

### 4.1. Benchmarking Against Baseline Models

In order to validate our findings, we conducted a comprehensive performance comparison of GramSeq-DTA, which now includes integrated RNA-Seq information. We compared it against several well-established baseline models as follows: DeepDTA, GraphDTA, DeepNC, and G-K-BertDTA. We used the MSE and CI values to assess performance. The model performance was compared across three benchmark datasets—BindingDB, Davis, and KIBA. DeepDTA employs a deep learning framework to capture the complex features of drug–target interactions using convolutional neural networks. GraphDTA, on the other hand, leverages graph neural networks to represent molecular structures as graphs, enabling it to better capture the topological properties of molecules. DeepNC uses a neural collaborative filtering approach to model interactions, focusing on latent feature extraction. Lastly, G-K-BertDTA integrates graph-based representations with BERT-like architectures to enhance the contextual understanding of molecular relationships. Our extensive evaluation, conducted on processed benchmark datasets, showed that GramSeq-DTA consistently outperformed its counterparts in terms of Concordance Index (CI) values, a widely accepted metric for evaluating predictive performance in drug–target affinity modeling. In the BindingDB dataset, GramSeq-DTA outperforms G-K-BertDTA by 1.32% in terms of CI value. Similarly, on the Davis dataset, GramSeq-DTA shows a 0.89% advantage over DeepNC. On the KIBA dataset, GramSeq-DTA exceeds G-K-BertDTA by 2.75%. Importantly, the integration of RNA-Seq data into GramSeq-DTA provided valuable insights into gene expression patterns, contributing to the improved accuracy of the models in predicting drug–target interactions. Detailed results of this comparison can be found in Table 3, Table 4 and Table 5, where the enhanced GramSeq-DTA model demonstrates its robust performance, setting a new standard in the field.

### 4.2. Advantage of Integrating RNA-Seq Information

Table 6 indicates that when validated on the processed BindingDB dataset, GramSeq-DTA performs better, with a CI value of 0.843 when integrating RNA-Seq information compared to not integrating RNA-Seq information. Similar results are evident in Table 7 and Table 8, where validation is performed on the Davis and KIBA datasets, respectively. CI values of 0.796 and 0.708 are observed for the processed Davis and KIBA datasets, respectively, when RNA-Seq information is integrated. These observations prove that integrating RNA-Seq information with the corresponding drug and target structural information can enhance the drug–target affinity prediction ability of the model.

### 4.3. Performance on Original and Processed Data

Table 9 presents the comparative evaluation of the performance of the proposed model on the original benchmark datasets and the processed benchmark datasets. The results of the model integration with RNA-Seq information are shown for the processed datasets. As shown in Table 1, the number of interactions between the original and the processed datasets differs. Despite losing approximately 80% of data in processing, our model performs better on the processed BindingDB dataset, with a best CI value of 0.843, while for the original BindingDB dataset, the optimum CI value was 0.818. The results on the original and processed Davis dataset are also competitive. When processing the Davis dataset, we lost around 84% of data. Our model indicates a CI value of 0.809 for the original Davis dataset, while for the processed Davis dataset, the CI value is 0.796. Data loss during the processing of the KIBA dataset is 99.5%, which is the highest value among all three datasets. For the KIBA dataset, there is a significant difference in CI values (original: 0.823; processed: 0.708) for original and processed datasets. The performance difference for the KIBA dataset can be due to the excessive data loss while processing the original dataset. This data loss leads to reduced data diversity and thus affects the overall generalizability of the model while training. Based on the performance of the other two process datasets (BindingDB and Davis), with more data available for the processed KIBA dataset, there is a high possibility of achieving better performance.

## 5. Conclusions and Future Directions

In this study, we have demonstrated that incorporating chemical perturbation information can enhance drug–target affinity prediction. The core contribution of this research is in its transformation of data from a chemical perturbation assay and utilizing them as an extra modality along with drug and protein structural information. In this work, we transformed the raw chemical perturbation assay data from the L1000 project, a project where a vast array of chemical assays have been performed, and updates to the assays are still ongoing. This research could guide future work on better understanding affinity prediction among biological entities in the absence of the three-dimensional structural information of those entities.

Nevertheless, a fundamental limitation of this research is that information from a chemical perturbation assay may not be available for every drug in the widely used drug–target affinity benchmark datasets. As new drugs are being introduced as potent antidotes for various diseases, corresponding chemical perturbation studies need to be performed on a large scale. In addition, some drugs are also being discarded and prohibited for conventional usage due to the lack of efficacy and severe side effects. Another aspect to consider is the imbalance between active and inactive compounds in a dataset. Chemical perturbation data might help by adding meaningful biological signals, but they could also worsen things if they introduce too much noise. Processed datasets can reduce the imbalance by filtering out redundant inactive data. However, if not handled carefully, they might remove important active compounds or introduce biases that do not reflect real-world data. Future studies should investigate different approaches for transforming chemical perturbation information.

Recent studies have shown that representing drugs as graphs provides a better understanding of the positional aspects of interacting atoms within the compounds, providing deeper insights into the structure–activity relationship (SAR) of the compounds. In addition, the recent availability of predicted 3D structures for the target proteins provides an opportunity to integrate these 3D structural data into the model, therefore enhancing the prediction capabilities of the computational models. This integration can provide deeper insights into the relationship between the structure and function of the proteins, providing a better understanding of structure-based predictive DTA modeling. Moreover, introducing advanced feature extraction methods from biological entities can enhance prediction.

With the continuous developments in natural language-based approaches in deep learning, drug discovery and drug repurposing are fields where the application of such methodologies are becoming popular. As the drugs and proteins are represented as strings, the utilization of natural language-based approaches can enhance the prediction abilities of the computational models. In the future, a conjugation of the existing structure-based approaches and natural language-based approaches can aid in improving the model development for DTA and the related tasks in drug discovery.

In summary, this work underscores the importance of integrating additional data modalities in drug–target affinity prediction.

## Figures and Tables

**Figure 1 biomolecules-15-00405-f001:**
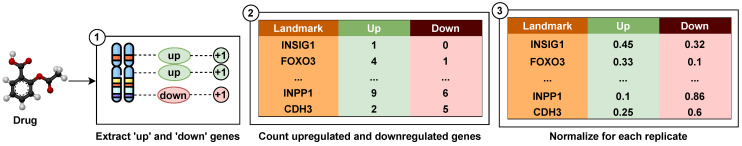
Preparation of the gene expression dataset. Gene expression information analyzed on 978 landmark genes for the selected drugs is extracted from the L1000 chemical perturbation data. After considering all the biological replicates of the perturbation analysis, a gene regulation matrix is created for both up-regulated and down-regulated genes.

**Figure 2 biomolecules-15-00405-f002:**
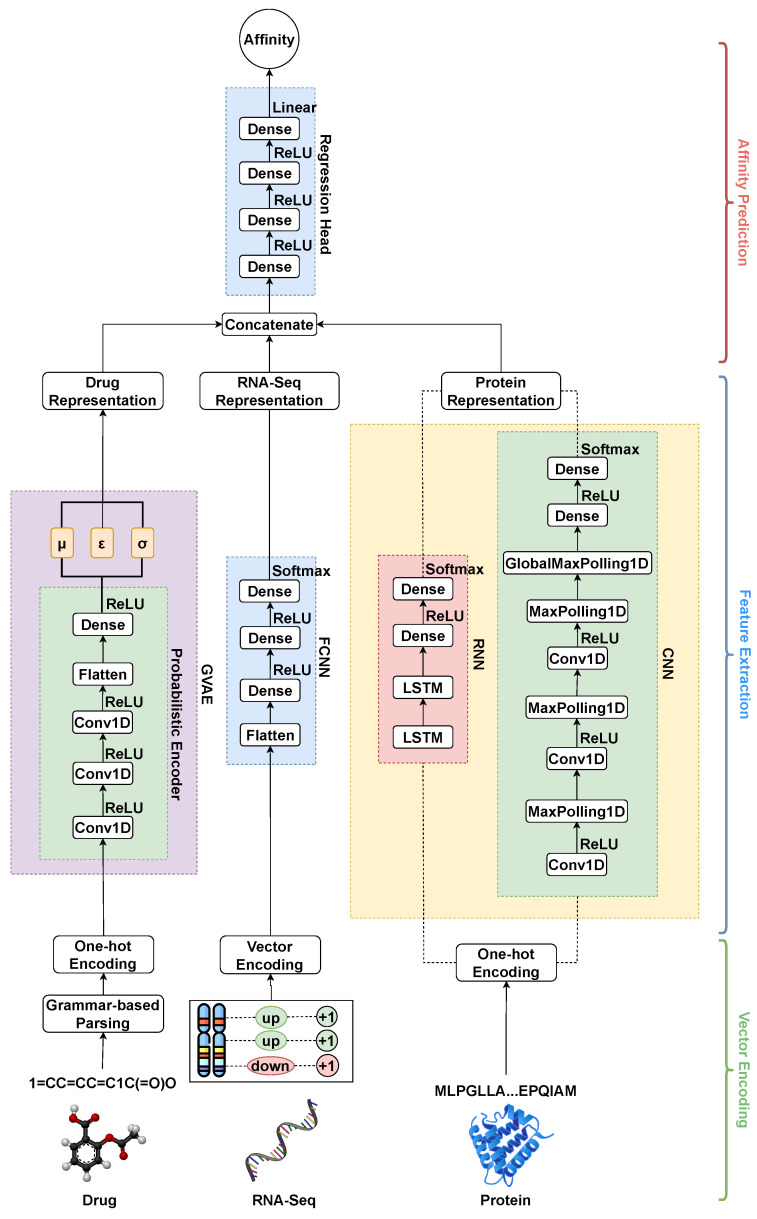
Network architecture of the proposed model. The encoded drug information is passed through an GVAE block, the RNA-Seq information is passed through an FCNN, while the encoded protein information is passed through a series of LSTM layers and 1D CNN layers. Learned representations are concatenated and passed through a FCNN acting as a regression head to predict the affinity.

**Figure 3 biomolecules-15-00405-f003:**
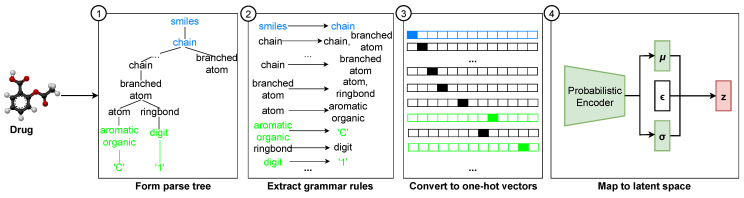
Encoding of drug SMILES structures. A parse tree is constructed based on the structural components of SMILES representations. Grammar rules are extracted from the parsed trees. SMILES representations are then converted into one-hot vectors. Finally, the one-hot vectors are transformed into the corresponding latent space representations using an encoder network.

**Table 1 biomolecules-15-00405-t001:** Dataset statistics—number of compounds, proteins, and interactions.

Dataset	Compounds	Proteins	Interactions
Original			
BindingDB	22,381	1860	91,751
Davis	68	379	30,056
KIBA	2068	229	118,254
Processed			
BindingDB	444	754	18,567
Davis	11	379	4169
KIBA	12	194	538

**Table 2 biomolecules-15-00405-t002:** Summary of network architectures and training hyperparameters.

Parameters	Value
Drug Encoding	
GVAE Encoder Filter Sizes	9, 9, 10
GVAE Encoder Kernel Sizes	9, 9, 11
GVAE Latent Space Dimension	56
RNA-Seq Encoding	
Dense Layers	2
Protein Encoding	
CNN Filter Sizes	32, 64, 96
CNN Kernel Sizes	4, 8, 12
RNN LSTM Layers	2
Regression Head	
Dense Layers	3
Training	
Epochs	500
Learning Rate	0.001
Batch Size	256
Optimizer	Adam

**Table 3 biomolecules-15-00405-t003:** Performance comparison of GramSeq-DTA with baseline models on the processed BindingDB dataset.

Model	Drug	RNA-Seq	Protein	MSE	CI
	Encoder	Encoder	Encoder		
DeepDTA	CNN	-	CNN	0.384	0.821
GraphDTA	GINConvNet	-	CNN	0.355	0.819
	GCNNet	-	CNN	0.397	0.786
	GATNet	-	CNN	0.512	0.757
	GAT_GCN	-	CNN	0.384	0.806
DeepNC	GENConv	-	CNN	0.367	0.828
G-K-BertDTA	GINConvNet + Embeddings	-	CNN	0.325	0.832
GramSeq-DTA	GVAE	FCNN	CNN	0.365	0.843
	GVAE	FCNN	RNN	0.355	0.832

**Table 4 biomolecules-15-00405-t004:** Performance comparison of GramSeq-DTA with baseline models on the processed Davis dataset.

Model	Drug	RNA-Seq	Protein	MSE	CI
	Encoder	Encoder	Encoder		
DeepDTA	CNN	-	CNN	0.219	0.779
GraphDTA	GINConvNet	-	CNN	0.187	0.771
	GCNNet	-	CNN	0.214	0.732
	GATNet	-	CNN	0.241	0.713
	GAT_GCN	-	CNN	0.227	0.753
DeepNC	GENConv	-	CNN	0.198	0.789
G-K-BertDTA	GINConvNet + Embeddings	-	CNN	0.169	0.778
GramSeq-DTA	GVAE	FCNN	CNN	0.293	0.796
	GVAE	FCNN	RNN	0.261	0.796

**Table 5 biomolecules-15-00405-t005:** Performance comparison of GramSeq-DTA with baseline models on the processed KIBA dataset.

Model	Drug	RNA-Seq	Protein	MSE	CI
	Encoder	Encoder	Encoder		
DeepDTA	CNN	-	CNN	0.877	0.609
GraphDTA	GINConvNet	-	CNN	1.061	0.628
	GCNNet	-	CNN	0.903	0.631
	GATNet	-	CNN	0.957	0.609
	GAT_GCN	-	CNN	0.831	0.671
DeepNC	GENConv	-	CNN	0.769	0.648
G-K-BertDTA	GINConvNet + Embeddings	-	CNN	0.693	0.689
GramSeq-DTA	GVAE	FCNN	CNN	0.843	0.708
	GVAE	FCNN	RNN	1.269	0.688

**Table 6 biomolecules-15-00405-t006:** Performance comparison of GramSeq-DTA with and without RNA-Seq information integration on the processed BindingDB dataset.

Model	Drug	RNA-Seq	Protein	MSE	CI
	Encoder	Encoder	Encoder		
GramSeq-DTA	GVAE	-	CNN	0.495	0.746
	GVAE	-	RNN	0.495	0.754
	GVAE	FCNN	CNN	0.365	0.843
	GVAE	FCNN	RNN	0.355	0.832

**Table 7 biomolecules-15-00405-t007:** Performance comparison of GramSeq-DTA with and without RNA-Seq information integration on the processed Davis dataset.

Model	Drug	RNA-Seq	Protein	MSE	CI
	Encoder	Encoder	Encoder		
GramSeq-DTA	GVAE	-	CNN	0.277	0.705
	GVAE	-	RNN	0.311	0.716
	GVAE	FCNN	CNN	0.293	0.796
	GVAE	FCNN	RNN	0.261	0.796

**Table 8 biomolecules-15-00405-t008:** Performance comparison of GramSeq-DTA with and without RNA-Seq information integration on the processed KIBA dataset.

Model	Drug	RNA-Seq	Protein	MSE	CI
	Encoder	Encoder	Encoder		
GramSeq-DTA	GVAE	-	CNN	1.011	0.653
	GVAE	-	RNN	0.876	0.618
	GVAE	FCNN	CNN	0.843	0.708
	GVAE	FCNN	RNN	1.269	0.688

**Table 9 biomolecules-15-00405-t009:** Performance of GramSeq-DTA without RNA-Seq information integration on the original benchmark datasets.

Dataset	Model	Drug	RNA-Seq	Protein	MSE	CI
		Encoder	Encoder	Encoder		
BindingDB						
Original	GramSeq-DTA	GVAE	-	CNN	1.029	0.818
		GVAE	-	RNN	1.061	0.812
Processed	GramSeq-DTA	GVAE	FCNN	CNN	0.365	0.843
		GVAE	FCNN	RNN	0.355	0.832
Davis						
Original	GramSeq-DTA	GVAE	-	CNN	0.446	0.806
		GVAE	-	RNN	0.445	0.809
Processed	GramSeq-DTA	GVAE	FCNN	CNN	0.293	0.796
		GVAE	FCNN	RNN	0.261	0.796
KIBA						
Original	GramSeq-DTA	GVAE	-	CNN	0.272	0.823
		GVAE	-	RNN	0.277	0.823
Processed	GramSeq-DTA	GVAE	FCNN	CNN	0.843	0.708
		GVAE	FCNN	RNN	1.269	0.688

## Data Availability

The data that support the findings of our study can be found in the GitHub repository, https://github.com/debnathk/gramseq.git (accessed on 16 January 2025).

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
