# Peer review of "GramSeq-DTA: A Grammar-Based Drug–Target Affinity Prediction Approach Fusing Gene Expression Information"

_biomolecules, 2025, doi:10.3390/biom15030405_

Round 1

Reviewer 1 Report

Comments and Suggestions for Authors

The paper presents GramSeq-DTA, a novel deep learning-based drug-target affinity (DTA) prediction method that integrates molecular structure and gene expression data. The model enhances predictive accuracy over benchmark datasets by employing a Grammar Variational Autoencoder (GVAE) for drug encoding, CNNs/RNNs for protein representation, and L1000 gene expression data (BindingDB, Davis, KIBA). The authors demonstrate that incorporating functional drug responses improves DTA performance compared to existing models like DeepDTA and GraphDTA. The study provides a compelling case for integrating structural and functional modalities in DTA prediction, with the potential for further enhancements via 3D structural data and NLP-driven feature extraction. I suggest minor revisions to clarify specific methodological choices and enhance result interpretations, but overall, the study is strong and suitable for acceptance.

  1. Introduction and Background - The manuscript presents GVAE as a method to encode drug structures while capturing their syntactic and semantic properties. However, given that GVAE operates on parse tree representations, how does the model handle chemically invalid molecules that might still be syntactically valid? Is there any post-processing step to filter out these invalid molecules before incorporating them into the DTA prediction framework?
  2. Introduction and Background - Integrating L1000 gene expression perturbation data is an interesting addition to capturing the functional characteristics of drugs. However, gene expression profiles can be highly context-dependent, varying across different cell types and conditions. How does the model account for this variability when integrating these features into drug-target affinity prediction? Would normalizing expression data across multiple conditions improve robustness?
  3. Materials and Methods - The study utilizes three widely used DTA datasets (BindingDB, Davis, and KIBA) with different characteristics, particularly regarding the number of drug-target interactions and label definitions. Given that these datasets differ in composition and scale, what were the specific criteria for selecting them? How does the model's performance vary across these datasets, and what insights can be drawn about its generalizability?
  4. Materials and Methods - The L1000 dataset is processed by selecting perturbagens at a standardized concentration (10 μM) and normalizing gene regulatory values. However, gene expression responses can be dose-dependent and vary significantly across different experimental conditions. How does this approach account for potential biases introduced by excluding data at other concentrations? Would including a range of concentrations improve model robustness, or does the single-dose selection provide a sufficient representation of functional drug effects?
  5. Results and Discussion / Conclusion - The study demonstrates that integrating RNA-Seq information improves the predictive accuracy of the GramSeq-DTA model across all datasets. However, RNA-Seq data availability is often limited for many drugs, as acknowledged in the study's limitations. How does the model handle missing RNA-Seq information, and would it still be beneficial for datasets where such data is sparse or unavailable?
  6. Results and Discussion / Conclusion - The study reports a significant reduction in interactions after dataset processing, with data loss reaching up to 99.5% in the KIBA dataset. Given the large discrepancy in data availability before and after processing, how does this affect the model's generalizability? Could the observed drop in CI values for the processed KIBA dataset be attributed to reduced data diversity, and how would the model perform with alternative processing strategies that retain more interactions?

Author Response

Comments 1: Introduction and Background - The manuscript presents GVAE as a method to encode drug structures while capturing their syntactic and semantic properties. However, given that GVAE operates on parse tree representations, how does the model handle chemically invalid molecules that might still be syntactically valid? Is there any post-processing step to filter out these invalid molecules before incorporating them into the DTA prediction framework?

Response 1: Thank you for your question. All the molecules present in the benchmark datasets are clinically tested against certain target proteins; thus, there is no possibility of encountering chemically invalid molecules. Nevertheless, human errors may occur while documenting the compound SMILES, which can make certain molecules chemically invalid. To deal with that, the RDKit framework is used to preprocess SMILES before passing them onto parse construction and screening those molecules. RDKit is also used to canonize all the SMILES to a standardized representation of the SMILES, as the same molecule collected from different sources may possess different SMILES representation. (More explanation added in Section 3.2.1, Lines 223-225).

Comments 2: Introduction and Background - Integrating L1000 gene expression perturbation data is an interesting addition to capturing the functional characteristics of drugs. However, gene expression profiles can be highly context-dependent, varying across different cell types and conditions. How does the model account for this variability when integrating these features into drug-target affinity prediction? Would normalizing expression data across multiple conditions improve robustness?

Response 2: For this work, upregulated and downregulated genes with a sample concentration of 10 uM are selected; other instances are discarded. The reason behind selecting 10 uM as our preferred sample concentration is that samples with 10 uM represent the most frequent ones when performing a sample concentration analysis on all the samples. Furthermore, we normalized the expression data for multiple replicates. Considering multiple conditions does not correspond to this work (Discussed in Section 3.1.2, Lines 188-192).

Comments 3: Materials and Methods - The study utilizes three widely used DTA datasets (BindingDB, Davis, and KIBA) with different characteristics, particularly regarding the number of drug-target interactions and label definitions. Given that these datasets differ in composition and scale, what were the specific criteria for selecting them? How does the model's performance vary across these datasets, and what insights can be drawn about its generalizability?

Response 3: BindingDB, Davis, and KIBA are some of the widely used datasets for benchmarking DTA. BindingDB contains a wide range of drug-target interactions; Davis consists of kinases and protein targets and corresponding kinase inhibitors with Kd values as labels; and KIBA conjugates multiple affinity metrics into a unified KIBA score. These differences, thus, allow a comprehensive evaluation of the robustness and generalizability of the model. The model’s performance across various datasets can vary. More comprehensive results can be observed in domain-specific datasets like Davis and KIBA, while the diversity of the BindingDB dataset may cause challenges to the model (More on this discussed in Section 4.3).

Comments 4: Materials and Methods - The L1000 dataset is processed by selecting perturbagens at a standardized concentration (10 μM) and normalizing gene regulatory values. However, gene expression responses can be dose-dependent and vary significantly across different experimental conditions. How does this approach account for potential biases introduced by excluding data at other concentrations? Would including a range of concentrations improve model robustness, or does the single-dose selection provide a sufficient representation of functional drug effects?

Response 4: For high-throughput screening, selecting a standardized concentration of 10 uM out of all other concentrations in the L1000 datasets offers consistency across perturbagens while simplifying pre-processing of the data. Indeed, gene expression responses can be dose-dependent, so selecting single-dose samples provides more biological relevance, providing a standardized functional overview of drug effects. While considering diverse concentrations could capture more nuanced dose-response relationships, it may increase the complexity. Thus, a single-dose strategy best suits this particular use case (More on this discussed in Section 3.1.2, Lines 188-192).

Comments 5: Results and Discussion / Conclusion - The study demonstrates that integrating RNA-Seq information improves the predictive accuracy of the GramSeq-DTA model across all datasets. However, RNA-Seq data availability is often limited for many drugs, as acknowledged in the study's limitations. How does the model handle missing RNA-Seq information, and would it still be beneficial for datasets where such data is sparse or unavailable?

Response 5: As RNA-Seq information is not present for all the drugs in each dataset, only those drugs are selected for evaluation whose corresponding RNA-Seq data is available. As per the scope of this work, there are no techniques applied to fill-in missing RNA-Seq data. We can rely on availability of experimental RNA-Seq readings in public databases, and we believe that more availability of RNA-Seq information would enhance the performance of our model (More on this discussed in Sections 3.1.2, Lines 207-209 and 5, Lines 365-367).

Comments 6: Results and Discussion / Conclusion - The study reports a significant reduction in interactions after dataset processing, with data loss reaching up to 99.5% in the KIBA dataset. Given the large discrepancy in data availability before and after processing, how does this affect the model's generalizability? Could the observed drop in CI values for the processed KIBA dataset be attributed to reduced data diversity, and how would the model perform with alternative processing strategies that retain more interactions?

Response 6: Thanks for pointing this out. The data loss for all the datasets, including KIBA, is indeed one of the major limitations of this work. The data loss surely affects the generalizability of the model while training. Designing a method to handle the data loss caused by the unavailability of the RNA-Seq data is our future goal (More on this discussed in Section 4.3 and Section 5).

Reviewer 2 Report

Comments and Suggestions for Authors

Overall, this is a rather vague and confusing paper. It is not clear how the authors developed GramSeq-DTA. Details and essential steps are missing.

It is not clear what the authors mean by drug-target affinity. The cited references use different definitions, from decades old chemical binding affinity to biological activity that may capture the desired outcome to predict drug-target interaction that affect a biological outcome. Given that the authors use the L1000 dataset of gene-signatures between drug unperturbed and perturbed states, I presume the authors try to achieve the latter. However, their benchmarking datasets only capture binding - which, of course, is often a necessary but certainly not a sufficient condition to induce a biological activity. Their benchmarking datasets are rather restrictive in this respect. As the name implies, BindingDB only includes drug-target binding information. Davis and KIBA are based on drug-based kinase inhibition. Davis focuses on drug-target specificity. Only KIBA tries to include different metrics for biological activity, e.g. such as ID50, Kd or Ki measures.

With respect to the different encoders; the authors use a pretrained grammar variational autoencoder (GVAE) for drug encoding. Everything else is rather vaguely defined. What are the details of the neural networks used for RNAseq and protein encoding? What is the source and number of proteins used in the protein encoder, e.g. for any training?

The number of compounds, proteins and interactions from the three benchmark datasets used pales in comparison with available data. For example, in ChEMBL there are 2.5 million compounds available, PDB stores about 240.000 protein structures and over 800,000 interactions have been identified between proteins and ligands. Certainly, only a representative sample of compounds, RNA sequences and proteins is necessary for performance testing. However, not only are the databases rather restrictive on specific protein classes, such as Davis and KIBA on kinases. Also, there is a discrepancy on compounds and proteins available in the corresponding databases and used in the manuscript. For example, BindingDB includes 20 times more compounds than were used by the authors. So, what compounds have the authors used for performance evaluation?

Have the different encoders pre-trained separately, or combined in the network model shown in Fig. 2? How have they been trained? What training set was used? The 6 lines in section 3.3 are insufficient to explain the training. The parameter values in table 2 are meaningless without additional details.

What specifically are the values y that are tested by MSE?

Author Response

Comments 1: It is not clear what the authors mean by drug-target affinity. The cited references use different definitions, from decades old chemical binding affinity to biological activity that may capture the desired outcome to predict drug-target interaction that affect a biological outcome. Given that the authors use the L1000 dataset of gene-signatures between drug unperturbed and perturbed states, I presume the authors try to achieve the latter. However, their benchmarking datasets only capture binding - which, of course, is often a necessary but certainly not a sufficient condition to induce a biological activity. Their benchmarking datasets are rather restrictive in this respect. As the name implies, BindingDB only includes drug-target binding information. Davis and KIBA are based on drug-based kinase inhibition. Davis focuses on drug-target specificity. Only KIBA tries to include different metrics for biological activity, e.g. such as ID50, Kd or Ki measures.

Response 1: Thank you for your question. For the BindingDB and Davis datasets, we used the Kd value (dissociation constant) for a drug-target pair as the measure of their binding. Similarly, KIBA provides a unified score combining all the interaction metrics. Affinity essentially reflects the likelihood of interaction of a drug-target pair. In simple terms, a drug only functions when it binds to a target; the effect can be stimulatory or inhibitory later on. In this work, we only focus on the likelihood of interaction; the aftereffect is beyond the scope of this work (More on this discussed in Sections 1, Lines 26-29, and 3.1.1).

Comments 2: With respect to the different encoders; the authors use a pretrained grammar variational autoencoder (GVAE) for drug encoding. Everything else is rather vaguely defined. What are the details of the neural networks used for RNAseq and protein encoding? What is the source and number of proteins used in the protein encoder, e.g. for any training?

Response 2: For RNA-Seq and protein encoding, we are not using any pre-trained model, but rather blocks of neural network architectures. For RNA-Seq, we are using fully connected neural networks (FCNN) and for proteins, we are using convolutional neural networks (CNNs) and Recurrent neural networks (RNNs). The protein encoder is using only the proteins in the training dataset (Discussed in detail in Section 3.2 and Figure 2).

Comments 3: The number of compounds, proteins and interactions from the three benchmark datasets used pales in comparison with available data. For example, in ChEMBL there are 2.5 million compounds available, PDB stores about 240.000 protein structures and over 800,000 interactions have been identified between proteins and ligands. Certainly, only a representative sample of compounds, RNA sequences and proteins is necessary for performance testing. However, not only are the databases rather restrictive on specific protein classes, such as Davis and KIBA on kinases. Also, there is a discrepancy on compounds and proteins available in the corresponding databases and used in the manuscript. For example, BindingDB includes 20 times more compounds than were used by the authors. So, what compounds have the authors used for performance evaluation? - Databases like PubChem, ChEMBL contain information for millions of compounds, PDB and now AlphaFold is a dependable source for protein structures - what they often lack is possible compound-target interaction information. The three benchmark datasets used in this work.

Response 3: Thank you for pointing this out. BindingDB, Davis, and KIBA are some of the widely used datasets used for benchmarking of drug-target affinity prediction tasks (examples include the models used as baselines). While Davis and KIBA are readily usable, the BindingDB dataset often needs thorough preprocessing to eliminate the null information and keep data where the SMILES representation of drugs, amino acid sequence information of proteins, and Kd values of a drug-target pair are present. This preprocessing leads to shrinking of the dataset (More on this discussed in Section 3.1.1).

Comments 4: Have the different encoders pre-trained separately, or combined in the network model shown in Fig. 2? How have they been trained? What training set was used? The 6 lines in Section 3.3 are insufficient to explain the training. The parameter values in table 2 are meaningless without additional details.

Response 4: No pre-training is involved in this work; we used a model pre-trained on a large number of drugs as the drug encoder. The encoders for RNA-Seq and proteins are trained from scratch, as well as the regression head. The combined network is trained together in an end-to-end manner (More on this discussed in Section 3.3).

Comments 5: What specifically are the values y that are tested by MSE?

Response 5: The y values represent the negative log of Kd values for datasets BindingDB and Davis and the negative log of KIBA score for the KIBA dataset (More on this discussed in Section 3.4.1).

Reviewer 3 Report

Comments and Suggestions for Authors

Among the rapidly growing number of new deep learning procedures aimed at accurately predicting drug-target affinities (DTA), the authors attempt to integrate functional information (e.g., an RNA-seq encoder from the L1000 project chemical perturbation data) with drug structural information (e.g., a GVAE grammar-based autoencoder), further extracting the functional information of the targets using already known protein sequence encoders. According to the authors' findings, applying the new approach to commonly used DTA datasets confirms that using functional information such as gene expression can effectively help improve DTA prediction power.

The article chapters are clearly explained and well written, worthy of being published in the journal Biomolecules. Minor concerns are the following points:

- Line 54. Replace the "?" character by providing a reference
- Line 169 "in these datasets" appears twice
- Line 230 Add reference for FCNN encoder
- Lines 238-9 Add reference for CNN and RNN encoders
- Line 244 Change "all possible amino acid sequences" to "all possible amino acid symbols" or similar
- Line 248 Change "a vector" to "a tensor" or "a 3D array"

- Lines 269-274. Perhaps an explanation of why they prefer the concordance index to the Persson and especially to the Spearman coefficient would be helpful to the reader. Could the choice depend on the dataset used? Perhaps the Spearman coefficient could also be listed in Tables 4-9.

- The authors are free to leave Tables 6,7,8 as they are, however, in my opinion, for greater clarity, each of them should be merged with Tables 3,4,5 respectively and discussed accordingly.

- The imbalance between active and inactive compounds is a problem in DTA prediction. Please spend a few words to address the problem and its impact on rank-based evaluation metrics, and discuss how/if the problem can be alleviated/worsened when using gene expression data, as well as its incidence when using processed datasets instead of the original ones.

Author Response

Comments 1: Line 54. Replace the "?" character by providing a reference.

Response 1: Resolved in manuscript (Section 1, Line 55).

Comments 2: Line 169 "in these datasets" appears twice.

Response 2: Resolved in manuscript (Section 3.1.1, Line 169).

Comments 3: Line 230 Add reference for FCNN encoder.

Response 3: Resolved in manuscript (Section 3.2.2, Line 240).

Comments 4: Lines 238-9 Add reference for CNN and RNN encoders.

Response 4: Resolved in manuscript (Section 3.2.3, Line 250).

Comments 5: Line 244 Change "all possible amino acid sequences" to "all possible amino acid symbols" or similar.

Response 5: Resolved in manuscript (Section 3.2.3, Line 254).

Comments 6: Line 248 Change "a vector" to "a tensor" or "a 3D array".

Response 6: Resolved in manuscript (Section 3.2.3, Line 257).

Comments 7: Lines 269-274. Perhaps an explanation of why they prefer the concordance index to the Persson and especially to the Spearman coefficient would be helpful to the reader. Could the choice depend on the dataset used? Perhaps the Spearman coefficient could also be listed in Tables 4-9.

Response 7: As per the literature, C-index is preferred over spearman or Pearson correlation. The reason being the ability of C-index to measure how often a randomly chosen pair is ranked correctly with respect to the ground truth, which is particularly relevant for ranking-based problems like drug-target affinity prediction. Neither the spearman nor pearson coefficient is robust enough to handle the diversity of affinity values, thus less suitable compared to C-index (More on this discussed in Section 3.4.2).

Comments 8: The authors are free to leave Tables 6,7,8 as they are, however, in my opinion, for greater clarity, each of them should be merged with Tables 3,4,5 respectively and discussed accordingly.

Response 8: We kept the results from tables 3,4,5 from the ones in tables 6,7,8 because we wanted to show the performance of our model compared to baseline models (Tables 3,4,5), as well as when we integrate and don’t integrate RNA-Seq information (Tables 6,7,8) separately.

Comments 9: The imbalance between active and inactive compounds is a problem in DTA prediction. Please spend a few words to address the problem and its impact on rank-based evaluation metrics and discuss how/if the problem can be alleviated/worsened when using gene expression data, as well as its incidence when using processed datasets instead of the original ones.

Response 9: Thank you for pointing this out. Chemical perturbation data might help by adding meaningful biological signals, but it could also worsen things if it introduces too much noise. Processed datasets can reduce the imbalance by filtering out redundant, inactive data. However, if not handled carefully, they might remove important active compounds or introduce biases that do not reflect real-world data (More on this discussed in Section 5, Lines 370-377).

Round 2

Reviewer 2 Report

Comments and Suggestions for Authors

This is still a rather hard to read manuscript with insufficient detail. Although the application of the presented drug-affinity prediction algorithm is relevant for this journal, the manuscript is excessive in unexplained machine learning jargon. Furthermore, many details of their approach are still unclear. For example, what are the specific training sets used for drugs, RNAseq and proteins? What are the 972 landmark genes? As "Biomolecules" is not a computer science/machine learning journal, for example, what does "The combined network is trained together in an end to end manner" mean (line 269)? In particular, end-to-end deep learning comes with the challenge requiring a large amount of training data, which is not properly introduced in this manuscript.
Also, when comparing the different approaches of other DTA machine learning methods mentioned in the manuscript, Compared to the other methods, it seem that GramSeq-DTA is only an marginal improvement in approach and certainly performance.
With respect to the test sets. Although the used sets have been used in the past, there is no reason to continue to use them now, given their restrictions as mentioned previously (e.g. Davis and KIBA is restricted on kinases).

Thus, overall, I find this manuscript out of scope of this journal, drowning the reader in machine learning jargon and missing essential details.

Author Response

Comments 1: For example, what are the specific training sets used for drugs, RNAseq and proteins?

Response 1: Each of the benchmark datasets we used in this work is divided into three sets: the training set (to train the model by adjusting its parameters to learn patterns from the data), the validation set (to fine-tune model hyperparameters and evaluate model performance during training), and the test set (to evaluate the final model performance on the unseen data to get the generalizability of the model). Thus, the training set is a subset of the complete dataset.

Comments 2: What are the 972 landmark genes? 

Response 2: L1000 is a high-throughput gene expression assay that measures the mRNA transcript abundance of 978 "landmark" genes from human cells. In the L1000 assay, they concluded that 978 genes are sufficient to capture any cellular state as a reduced representation of the human transcriptome. Thus, they call this list of genes “landmark genes” (more on this discussed in Page 5, Line 207).

Comments 3: As "Biomolecules" is not a computer science/machine learning journal, for example, what does "The combined network is trained together in an end to end manner" mean (line 269)? In particular, end-to-end deep learning comes with the challenge requiring a large amount of training data, which is not properly introduced in this manuscript.

Response 3: In a machine learning context, end-to-end means training a model from start to finish to solve a particular task. For our work, the model starts with extracting features from the given data, then learns the pattern of inter-relations among the data from those extracted features, and finally predicts a continuous value as output. So, from extracting features to predicting an output, the complete process is called “end-to-end”. 

Every deep learning model training comes with the challenge of requiring a large amount of data for training. However, how much data is required may vary depending on the task in hand. Drug-target affinity prediction is indeed a complex problem to model, but as per the earlier studies, the data required for training is not huge. At the end, it is a regression task where we are predicting a continuous value. The model is extracting features from biological entities, learning patterns of interactions, and mapping those patterns with the provided labels. Thus, it can be emphasized that the amount of data we are using is sufficient for training the model. However, our analysis also suggests that our method's performance may improve further if we have more training data.

Comments 4: Also, when comparing the different approaches of other DTA machine learning methods mentioned in the manuscript, Compared to the other methods, it seem that GramSeq-DTA is only an marginal improvement in approach and certainly performance.

Response 4: In the area of AI/deep learning research, often such evaluation metrics are used, which basically give a continuous value ranging from 0 to 1. The reason for this is for easier visualization of the comparative results. Either it be a large language model (LLM) or a simple classification task, it goes for all. Even though the difference in performance may be in decimals, the underlying impact is much greater than that. Thus, we can conclude that our model beats the performance of baseline models, even if the difference is in decimals.

Comment 5: With respect to the test sets. Although the used sets have been used in the past, there is no reason to continue to use them now, given their restrictions as mentioned previously (e.g. Davis and KIBA is restricted on kinases).

Response 5: Thanks for pointing that out. Davis and KIBA contain kinase-proteins and corresponding inhibitors, but on the other hand, BindingDB, another benchmark dataset we used, is not restricted to kinases. These are some of the most widely used datasets for evaluating drug-target affinity prediction; that is why they are called "benchmarks.". As we needed to compare the performance of our model with the existing ones, we had to use these datasets.